# EFC-YOLO: An Efficient Surface-Defect-Detection Algorithm for Steel Strips

**DOI:** 10.3390/s23177619

**Published:** 2023-09-02

**Authors:** Yanshun Li, Shuobo Xu, Zhenfang Zhu, Peng Wang, Kefeng Li, Qiang He, Quanfeng Zheng

**Affiliations:** School of Information Science and Electrical Engineering, Shandong Jiaotong University, Jinan 250357, China; 21208021@stu.sdjtu.edu.cn (Y.L.); zhuzf@sdjtu.edu.cn (Z.Z.); 205049@sdjtu.edu.cn (P.W.); 205073@sdjtu.edu.cn (K.L.); 21208022@stu.sdjtu.edu.cn (Q.H.); 21208007@stu.sdjtu.edu.cn (Q.Z.)

**Keywords:** surface defect detection, YOLOv7, deep learning, feature extraction

## Abstract

The pursuit of higher recognition accuracy and speed with smaller model sizes has been a major research topic in the detection of surface defects in steel. In this paper, we propose an improved high-speed and high-precision Efficient Fusion Coordination network (EFC-YOLO) without increasing the model’s size. Since modifications to enhance feature extraction in shallow networks tend to affect the speed of model inference, in order to simultaneously ensure the accuracy and speed of detection, we add the improved Fusion-Faster module to the backbone network of YOLOv7. Partial Convolution (PConv) serves as the basic operator of the module, which strengthens the feature-extraction ability of shallow networks while maintaining speed. Additionally, we incorporate the Shortcut Coordinate Attention (SCA) mechanism to better capture the location information dependency, considering both lightweight design and accuracy. The de-weighted Bi-directional Feature Pyramid Network (BiFPN) structure used in the neck part of the network improves the original Path Aggregation Network (PANet)-like structure by adding step branches and reducing computations, achieving better feature fusion. In the experiments conducted on the NEU-DET dataset, the final model achieved an 85.9% mAP and decreased the GFLOPs by 60%, effectively balancing the model’s size with the accuracy and speed of detection.

## 1. Introduction

In modern industrial production, steel plays a crucial role in numerous fields due to its excellent characteristics. With the continuous increase in steel production, ensuring the quality of steel products becomes increasingly vital. During industrial manufacturing, metal products often exhibit various surface defects, such as cracks, pits, pores, burrs, and others. These defects can significantly impact the quality and performance of the products. Indeed, from the perspectives of ensuring product quality control, ensuring the safety of steel usage, and controlling production costs, the development of technologies for surface defect detection in steel is of paramount importance and in high demand [1].

Initially, defect detection heavily relied on manual labor, which exhibited limitations such as excessive dependence on the experience of the inspection personnel, high costs, and low efficiency. However, with the advent of electrification and automation control technologies, many tasks that were previously performed manually have gradually been taken over by automated equipment [2]. The emergence and advancement of computer technology [3] further propelled the progress of automated industries [4], thus creating an urgent demand for defect detection algorithms that not only meet automation requirements but also ensure high detection accuracy [5].

Object detection algorithms represent a significant research direction in the field of computer vision. Deep-learning-based object detection algorithms utilize convolutional neural networks (CNNs) to transform raw input information into more abstract and higher-dimensional features for learning. This feature-learning approach allows for the automatic fitting of the essential characteristics required for object detection and classification. The powerful feature-expression ability and generalization of higher-dimensional features also make them perform well in complex scenarios, so the use of object detection can meet most application requirements in the industry, including defect detection [6]. Currently, surface-defect-detection methods can be primarily classified into two categories: single-stage detection methods and two-stage detection methods. Single-stage methods include the You Only Look Once (YOLO) algorithm family [7,8,9,10,11], Single-Shot MultiBox Detector (SSD) [12], etc. The single-stage object detection algorithm is a method to predict the target position and category directly from the input image, and usually, only one forward propagation is needed to complete the object detection task. Representative methods of the two stages include the region-based convolutional neural network (R-CNN), Fast-RCNN, Faster-RCNN [13,14,15], and MASK RCNN [16]. The two-stage object detection algorithm can obtain more accurate detection results by generating candidate boxes and then classifying and fine-tuning the candidate boxes, compared with the single-stage object detection algorithm, but the operation speed is relatively slow due to complex operations.

YOLO is a single-stage object detection algorithm that prioritizes speed. It achieves this by dividing the input image into multiple grids and predicting the position, class, and confidence score of several bounding boxes for each grid. Therefore, YOLO has the advantages of strong real-time performance, high precision, and good scalability, which can help it better detect subtle defects on the steel surface. While earlier versions of the YOLO algorithm displayed significant advantages in real-time object detection and related aspects, they faced challenges when dealing with small objects and dense object arrangements. To address the aforementioned issues, the YOLO series of algorithms have employed strategies such as using more complex backbone networks, optimizing multi-scale feature fusion in the neck network, and adopting advanced loss functions. These approaches aim to improve the network’s ability to fit the data more effectively [17,18]. However, as the accuracy of algorithms continues to improve, there is a corresponding increase in both model runtime speed and size. In contrast, the YOLOv7 algorithm [11] incorporates advanced auxiliary detection heads and model-scaling strategies. It introduces the E-ELAN (extended efficient layer aggregation networks) module, which controls the longest and shortest gradient paths to aid deeper models in learning features and converging more effectively. And RepConv enhances the model’s feature-representation capability by learning additional parameters during the training phase. By incorporating RepConv before the detection heads of three different scales, this approach allows for a marginal sacrifice in training time while achieving faster inference speed and higher accuracy, ultimately enhancing the model’s performance. Therefore, this paper uses the YOLOv7 algorithm as the baseline to conduct experiments on the NEU surface defect database (NEU-DET) [19] to evaluate the model.

Computational cost is an essential factor that cannot be overlooked. When detection performance is comparable, a model with a lower computational cost is undoubtedly more desirable as it allows for compatibility with a broader range of devices. Therefore, the lightweight design concept has been a significant direction for model improvement, exemplified by networks such as MobileNets [20,21,22], ShuffleNets [23,24], and GhostNet [25]. These networks are built upon lightweight designs and utilize Depthwise Separable convolution [21] or Group Convolution (GConv) [26] to extract spatial features. While Depthwise Separable convolution significantly reduces the number of Floating Point Operations (FLOPs), its improvement in speed is not very substantial due to frequent memory access [27]. In response, FasterNet proposes a novel Partial Convolution (PConv) as an alternative. PConv reduces computation redundancy and the number of memory accesses to eliminate this disparity. Consequently, PConv more efficiently leverages the computational capabilities of the device. Therefore, we chose to improve our network based on FasterNet.

We propose the Efficient Fusion Coordination network (EFC-YOLO) for steel surface defect detection, which is based on the YOLOv7 algorithm serving as the foundational model. In the backbone network of EFC-YOLO, we employ the Fusion Faster block, which is based on the PConv as the fundamental operator. This design ensures high accuracy while reducing redundant computations and accelerating memory access. Compared to the original YOLOv7 backbone, the EFC-YOLO backbone is more precise, has a smaller algorithm size, and requires fewer computations. In addition, we propose a novel attention mechanism called Shortcut Coordinate Attention (SCA) mechanism. SCA captures long-range dependencies and adjacent information correlations more effectively by separately encoding horizontal and vertical directions. Furthermore, the incorporation of convolutional shortcuts allows for better attention to global dependency relationships, thereby addressing the limitations of the CA attention mechanism [28], which primarily focuses on local information and may overlook contextual relationships. Finally, we incorporate the Weighted Bi-directional Feature Pyramid Network (BiFPN) structure [29], which facilitates the full propagation of original feature information for feature reuse and enhances the diversity of robust feature information. By reducing the loss of feature information during multiple convolution and upsampling processes, this addition effectively improves detection accuracy. The main contributions are as follows:We utilize PConv as the primary operator and further propose the Fusion-Faster module based on the FasterNet module to be integrated into the backbone feature extraction network. With its compact size, the Fusion-Faster module demonstrates more efficient spatial information extraction capabilities while also strengthening the abilities of feature extraction and fusion;We propose the SCA module, which is a structural improvement based on the Coordinate Attention (CA) mechanism. By incorporating convolutional shortcuts, we aim to mitigate the negative impact of attention modules on inference speed. This enhancement enables the network to better capture long-range dependencies and contextual information, improving its overall performance;The BiFPN is used to reduce unnecessary branches, reduce the loss of feature information in the convolution process, and make deep features better multi-scale feature fusion to transmit more effective weight information. As a result, the network’s ability to capture diverse and essential features at different scales is significantly enhanced, ultimately improving the overall performance of the model.

## 2. Related Work

### 2.1. Traditional Machine Learning Methods

Early traditional computer-vision based methods for metal-defect-detection heavily relied on manually designed features. These methods typically involved various steps, such as image preprocessing, feature extraction, and defect detection and classification, to identify and analyze defects in metal materials. Some traditional feature extraction and classification algorithms include Local Binary Pattern (LBP) [30], Histogram of Oriented Gradients (HOG) [31], and Gray-Level Co-occurrence Matrix (GLCM) [32]. Zhang et al. [33] proposed a method that combines fitting the histogram and membership matrix of test images with Gaussian functions to locate and detect defects. Prasitmeeboon [34] uses color histograms and support vector machines to detect particleboard defects and threshold and smoothing techniques to locate defects. Zhao et al. [35] used vector-valued regular kernel approximation and support vector machine to locate defects in steel products. Traditional defect detection methods are not flexible enough for complex defect shapes and variations. A deep-learning-based approach can automatically learn feature representations to adapt to different defect shapes and sizes for more accurate and robust defect detection.

### 2.2. Deep Learning Object Detection Methods

In recent years, the rapid development and gradual popularization of deep learning has also begun to be widely used in the field of defect detection. Compared with traditional methods, the deep learning method does not need to manually extract features but automatically extracts features and fits models through automatic machine learning data-updating parameters and can automatically classify defect types and predict defect location after placing samples into the network.

There are many excellent defect detection algorithms that have been proposed based on the YOLO algorithm. For instance, Zhao et al. [36] introduced RDD-YOLO, which incorporates Res2Net blocks into the backbone network to enlarge the receptive field. Additionally, they redesigned the Dual-Feature Pyramid Network (DFPN) to enhance neck feature extraction and reuse lower-level features. Xie et al. [37] proposed the Feature-Enhanced YOLO (FE-YOLO) algorithm for surface defect detection. They combined Depthwise Separable convolutions and dense connections to reduce model size while maintaining performance and employed the Dual Feature Pyramid Network (DFPN) to enhance spatial positional relevance in multi-scale detection layers. Qian et al. [38] proposed the LFF-YOLO algorithm, which utilizes ShuffleNetv2 as the feature-extraction network to reduce the number of parameters. For the neck, a Lightweight Feature Pyramid Network (LFPN) is employed to improve the efficiency of multi-scale feature fusion. Finally, they introduced the Adaptively Receptive Field Feature Extraction (ARFFE) module, which uses weighted multi-receptive field channels to address the challenge of fixed receptive fields in adapting to defects of different scales. Yu et al. [39] introduced the Diagonal Feature Fusion Network (DFN) strategy in the backbone network, which matches multi-scale features without sacrificing speed. This strategy eliminates the bottom-up pathway and reduces the overall dimension of feature maps in the diagonal direction, leading to a significant reduction in model size. In the neck network, residual modules are used to suppress overfitting and prevent gradient vanishing. Moreover, they designed an adaptive localization loss function, which allows for flexible sensitivity adjustment based on defect size. The aforementioned algorithms have made significant contributions in reducing model size, improving recognition accuracy, and enhancing speed. However, to further improve detection accuracy while ensuring high-speed operation, we propose our EFC-YOLO network.

The YOLOv7 algorithm and its variants have demonstrated exceptional performance thanks to their commendable balance between speed and accuracy. Consequently, these models have been employed by numerous researchers across various domains to accomplish diverse tasks. For instance, Hussain et al. [40] introduced a non-invasive multi-class pallet racking detection framework, utilizing YOLOv7 as the core code for an autonomous rack inspection framework based on computer vision. Addressing the issue of scarce data, they proposed a domain-variance modeling mechanism to generate more representative samples. Leveraging the lightweight nature of the YOLOv7 model, the algorithm was successfully deployed on embedded devices in adjustable forklift racks, effectively covering the shelves near forklift operations. Li et al. [41] aimed to enhance the precision of identifying different types of defects in Printed Circuit-Board Assembly (PCBA) by strengthening the PCBA detection algorithm’s capability to handle environmental factors and multi-modal data. They introduced a network model based on YOLOv7 and designed prediction heads based on CA attention mechanism. This approach maintains low-cost computation while better capturing remotely dependent global correlations and focusing on regions of interest. Furthermore, they incorporated lower-level feedback propagation into the neck feature pyramid, enhancing multi-scale information extraction during the feature fusion phase. Lastly, they proposed the SEIoU loss function, which augments positional information sensitivity by adding directional loss, leading to a more efficient anchor box regression. Chen et al. [42] proposed a novel ship-detection model named CSD-YOLO to address the issues of missed detections and false identifications in complex environments for multi-scale ship recognition in Synthetic Aperture Radar (SAR) images. CSD-YOLO is built upon the YOLOv7 baseline model, incorporating the SAS-FEN module that combines ASPP and self-attention mechanisms. This module enhances the fusion of feature information across network layers, improving the model’s understanding of ships at different scales and efficiently capturing scattering information from small targets. Moreover, the integration of the SIoU loss function accurately measures the overlap between predicted and actual bounding boxes, enhancing the precision of multi-scale ship detection.

## 3. Proposed Method

The overall architecture of the network is illustrated in Figure 1. The backbone network first performs feature extraction and fuses channel information, ultimately generating three different sizes of feature maps. The SCA attention module is incorporated at the end of the backbone network to enhance the weights of the regions of interest. Subsequently, feature maps of different resolutions are sent to the neck feature-fusion pyramid for further processing. The pyramid structure emphasizes the flow of information between adjacent levels of feature maps, allowing the spatial semantic information of deep and shallow layers to continuously merge and be weighted. This enables the extraction of finer-grained features and facilitates better discrimination of various types of defects. Finally, three different sizes of feature maps are obtained, and the YOLO head is employed for defect detection. The integration of the Fusion-Faster module and the SCA attention mechanism enhances the overall performance of the detector by reducing the model size, accelerating model convergence, and achieving higher-quality defect localization and classification.

### 3.1. Backbone Network Based on Fusion-Faster Module

For deep learning networks in the visual domain, a substantial amount of convolutional operations are required to extract spatial information from images, which constitutes the main task of the network. However, these extensive convolutional operations impose high demands on the device’s memory and computational power. Therefore, to meet the requirements of embedded devices used in practical industrial production and to reduce the computational complexity and large parameter size during operations, achieving a network lightweight model, it is essential to improve the basic operators.

Reducing the model’s parameter size can be achieved through methods such as the use of Depthwise Separable convolution , as proposed in [20]. Depthwise Separable convolution is a key component of MobileNets, and it decomposes the standard convolution operation into two separate steps, Depthwise convolution and Pointwise convolution, resulting in more efficient computations. Depthwise convolution is performed independently on each channel, where each channel uses a separate convolution kernel for convolutional operations. This significantly reduces the computational costs. On the other hand, Pointwise convolution, also known as 1×1 convolution, involves element-wise multiplication and accumulation operations, which are used to map the results of Depthwise convolution to the desired output channel dimension. Due to the nature of Depthwise convolution, which performs independent convolution calculations on each input channel, it has the significant advantage of reducing the model’s parameter count and computational complexity. For a tensor with an input shape of H×W×C, the number of FLOPs for convolution using a kernel size of k×k can be calculated as
(1)$$h\times w\times k^{2}\times c^{2}$$

Using Depthwise convolution, the number of FLOPs is reduced to
(2)$$h\times w\times k^{2}\times c$$
which is significantly lower compared to regular convolution, due to the reduction in the number of convolution kernels. Secondly, Depthwise convolution can keep the spatial size of both the input and output unchanged, which is very useful for some tasks that need to retain spatial information, such as image segmentation and object detection.

However, we cannot simply replace conventional convolutions with Depthwise convolution. Although Depthwise convolution can significantly reduce the number of parameters, it does not guarantee that lower FLOPs will necessarily result in a higher computational speed of the model. In some cases, while Depthwise convolution greatly reduces FLOPs, it may increase the frequency of memory access during model execution, leading to less significant improvements in inference speed than expected based on numerical estimates. Especially in shallow networks, where the feature map size is relatively large, the corresponding computational cost will also be much higher than for deep networks. If we excessively use Depthwise convolution in shallow networks solely to reduce FLOPs, it can have a more negative impact on inference speed. Considering our goal of reducing model size and increasing recognition accuracy without compromising inference speed, we adopt a design philosophy that emphasizes achieving these objectives. We employ PConv [27] as the fundamental operator to replace traditional Depthwise convolution, aiming to optimize the convolutional computational costs while avoiding frequent memory access and reducing computation redundancy.

Therefore, this paper proposes the Fusion-Faster module with the design objective of efficiently extracting features while integrating semantic information among multiple channels. As shown in Figure 2, the Fusion-Faster module extends the FasterNet module by adding an additional shortcut path that performs a single convolution, batch normalization, and activation function operation. This allows the module to have two distinct paths to process the original feature map. The outputs from the two branches are then concatenated together, and a 1 × 1 convolution is used to fuse the weights of each branch, enhancing the network’s recognition performance. The FasterNet module utilizes residual PConv and channel-wise ’squeeze and excite’ operations to propagate information. The inclusion of the residual structure in FasterNet effectively addresses the issues of vanishing gradients and exploding gradients, resulting in better preservation and propagation of the feature distributions from before the PConv operations. The shortcut branch undergoes a 1 × 1 convolutional operation, which compresses the channel dimensions while preserving the original feature distribution of the image to the maximum extent. This process reduces computational complexity and matches the channel numbers, allowing the weights of different layers to learn more diverse features and introduce additional non-linear transformations to the network while retaining the original feature distribution of the image.

This reduction in dimensionality helps to decrease the overall parameters of the branch, thereby lowering the computational cost. The feature map with *c* channels is downsampled into two feature maps with the same channel size of $\frac{c}{2}$ each. Subsequently, PConv is applied to the feature map samples, where the PConv operation is only applied to a subset of input channels to extract spatial features, while the remaining channels remain unchanged without any operation. In most cases, we assume that consecutive feature maps from different channels exhibit minimal differences in their features, and the feature distributions among these channels are highly similar. Therefore, the PConv operation does not cause significant discrepancies between channels that have undergone the operation and those that have not. As a result, subsequent operations on channel dimensions can further enhance information exchange among different channels. Pconv utilizes a scaling factor “*r*” to control the ratio of channels participating in the operation. It performs a 3×3 convolution only on $c_{p}\left(c_{p}=\frac{c}{r}\right)$ channels, while the remaining $\left(1-\frac{c}{r}\right)$ channels are not involved in the computation. Finally, the output channels from Pconv and the remaining channels are concatenated together. Therefore, for an input tensor with shape H×W×C, the computational complexity of PConv is only
(3)$$h\times w\times k^{2}\times c_{p}^{2}$$
where $c_{p}$ represents the number of channels involved in the PConv operation. When the scaling factor ratio $r=\frac{c}{c_{p}}=\frac{1}{4}$, the FLOPs of PConv are reduced to only $\frac{1}{16}$ of the regular convolution. Furthermore, compared to regular convolution, PConv also exhibits reduced memory access, i.e.,
(4)$$h\times w\times 2c_{p}^{}+k^{2}\times c_{p}^{2}\approx h\times w\times 2c_{p}^{}$$

When $r=\frac{1}{4}$, the memory access of PConv is only $\frac{1}{4}$ of regular convolution. After the PConv operation, we utilize the Pointwise convolution to perform an operation of increasing and then decreasing the channel dimensions, aiming to avoid imbalanced weight distributions and ensure better data fusion among different channels. To fully leverage the information from each channel, we first increase the channel dimensions using the Pointwise convolution with a channel number of 2c to integrate data among channels. Subsequently, we shrink the channel dimensions back to *c*, matching the size of the feature maps and completing the transmission of channel information. The Pointwise convolution combines feature information from different feature maps while simultaneously modifying the channel count. The addition and concatenation of multiple feature maps at the pointwise level also allow the network to learn richer features. Despite the Pointwise convolution using a kernel with just one pixel, the incorporation of a non-linear transformation through activation functions aids in extracting more discriminative features. After incorporating the computational load of Pointwise convolutions, the total computational complexity of PConv and Pointwise convolutions is
(5)$$h\times w\times \left(k^{2}\times c_{p}^{2}+c^{2}\right)$$

Pointwise convolution reduce the size of convolutional kernels, thereby reducing the number of network parameters, which is particularly beneficial for achieving our lightweight network. Additionally, residual connections are used after Pointwise convolution. The residual structure is connected to the original feature map, avoiding data loss during the convolution operation and preserving the basic original features. This further emphasizes the sample features while shortening the gradient-propagation path. The subsequent feature vectors and the feature vectors that have passed through the convolutional shortcut are stacked together in the network’s width direction using concatenation. Then, a Pointwise convolution is applied to further fuse the features. Similarly, the 1×1 convolutional operation consists of a Pointwise convolution, a batch-normalization (BN) layer, and a SiLU activation function, where the BN layer is placed after the convolutional layer and before the activation function. The combination of the BN layer and the activation function enhances the network’s non-linearity, thereby improving the model’s learning capacity and generalization ability. This helps to prevent issues of overfitting and underfitting in the model. In conclusion, the proposed Fusion-Faster module achieves further extraction and enhancement of spatial semantic information without compromising feature distribution. Moreover, the subsequent stacking and convolution operations better integrate the entire module’s operations, leading to improved model learning capacity and detection accuracy.

### 3.2. Shortcut Coordinate Attention Module

To meet the fundamental requirement of high-speed and accurate operation, we propose the Shortcut Coordinate Attention (SCA) module based on a lightweight design, as shown in Figure 3. This module enhances the identification of target defects while minimizing computational burden. The CA attention mechanism [28] computes the similarity between input feature vectors to determine attention weights and then performs a weighted summation of the feature vectors based on the attention weights. Compared to other complex attention mechanisms, the CA attention mechanism has fewer parameters and computational costs, making it well suited for achieving our goal of algorithm lightweighting.

Conventional convolutional operations are effective in extracting spatial information within the same channel. However, the use of a 3×3 convolutional kernel limits its ability to extract global information. On the other hand, global pooling operations, such as average pooling or max pooling, can to some extent achieve recognition of overall information by computing the average or maximum values across the entire feature map. The CA attention mechanism can better capture the positional information of detection targets, thereby allocating more weight to regions of interest within the targets. We decompose the channel attention into two parallel 1D pooling operations to alleviate the loss of positional information caused by 2D global pooling. The attention mechanism performs global average pooling separately along the H (height) and W (width) dimensions of the feature map. This process extracts and compresses the horizontal and vertical information of the feature map, resulting in two feature matrices: one containing the original image’s horizontal information with dimensions H×1×C, and the other containing the vertical information with dimensions 1×W×C. These matrices contain long-range positional information and enable the searching of image information in both horizontal and vertical dimensions. Performing global pooling operations in both directions allows the CA attention module to capture long-range spatial dependencies in one spatial direction while retaining precise positional information in the other spatial direction. This is beneficial for the network to accurately locate the regions of interest throughout the entire spatial domain. The two feature matrices will undergo matrix transposition to match their dimensions, and then they will be concatenated to form a vector for further computation. From many perspectives, the CA attention mechanism also has some limitations. As it primarily focuses on local information, it may overlook global contextual relationships. However, there exists a certain correlation between nearby positions and adjacent channels in the information. So we replace the original 1×1 convolution with a 1×3 convolution after concatenating the two feature matrices. This modification allows the operation, which previously only facilitated information exchange between channels, to also integrate semantic information from adjacent channels and positions. As a result, it effectively establishes the dependency between neighboring positions, enabling a better reflection of the overall features of the defects. This enhances the attention mechanism’s capability to strengthen the weights for relevant targets. Additionally, by setting a scaling factor ‘*r*’, we perform channel squeezing with $\frac{1}{r}$ of the original number of convolutional kernels. This operation enables the mutual fusion of spatial position information extracted in the previous step. As the number of channels involved in the computation decreases, the module’s computational complexity and parameter count are also reduced, leading to an improvement in inference speed. Subsequently, the vector containing horizontal and vertical information is split into two vectors, $f^{h}$ and ;$f^{w}$. Then, two 1×1 convolutions, $F_{h}$ and $F_{w}$, are employed to transform $f^{h}$ and $f^{w}$ into tensors $g^{h}$ and $g^{w}$ with the same number of channels as the input *X*.
(6)$$g^{h}=\sigma \left(F_{h}\left(f^{h}\right)\right)$$
(7)$$g^{w}=\sigma \left(F_{w}\left(f^{w}\right)\right)$$

The sigmoid function σ is used to enhance the model’s nonlinearity and better fit the model. Next, the attention sequence containing position information is multiplied with the original image to strengthen the focus on the positional information. Through experiments, we found that the additional computations introduced by the attention mechanism often slow down the inference speed. To ensure real-time detection, we applied an attention mechanism to only half of the channels, while the remaining channels were processed using a simple 1×1 convolutional shortcut to reduce the computational cost. Finally, the two parts are concatenated along the channel dimension.

Unlike the spatial attention mechanism, which directly assigns weights to the entire feature map, and the channel attention mechanism, which only redistributes the weights of different channels, our SCA module can focus more on different directions of spatial information and independently encode them. The attention information in the horizontal and vertical directions can be accurately captured and applied to the input feature map tensor. The attention-encoded information in the horizontal and vertical directions is also precisely identified through pooling operations to determine whether the relevant information of interest exists in the corresponding rows and columns. Compared to traditional attention mechanisms, the independent horizontal and vertical encoding processes in our SCA module allow for more accurate localization of objects of interest, thus helping the entire model to better locate and recognize targets. The Convolutional Shortcut branch reduces computational complexity while enhancing the expression ability of lower-level features, extracting richer features, and better adapting to the network’s requirements. The improvement using a 1×3 convolution kernel can pay more attention to the correlation of adjacent information. By applying the attention mechanism only to a subset of channels, we can ensure high-speed inference while maintaining recognition accuracy.

### 3.3. Neck Network Based on a Bi-Directional Feature Pyramid Network

In common recognition networks, the final recognition results are often determined by information from the deep parts of the backbone network. For a feature-extraction neural network, shallow-level feature maps generally contain more fine-grained semantic information and detailed features compared to deep-level feature maps. However, as the downsampling increases, deep-level feature maps may lose some fine details. The introduction of Feature Pyramid Networks (FPN) [43] effectively addresses these issues. In YOLOv7, a similar Path Aggregation Network (PANet) [44] is used as shown in Figure 4. It not only receives the output from the deepest layer of the backbone network but also incorporates additional semantic information from feature maps at different depths of the backbone network. The bidirectional pyramid feature-fusion structure processes feature maps with three different scales, obtained through multiple max pooling and downsampling operations, and then sends them to the neck network. In the neck network, upsampling is performed on small-sized images to enlarge the feature map while retaining deep-level information, thus matching the size of the feature map from the previous shallow-layer. The upsampled deep-level features and the original shallow-level features are stacked together along the channel dimension using the concat operation. Finally, the extended efficient layer aggregation network (ELAN) is used to shuffle and combine the channel information together. The three sizes of feature maps undergo multiple upsampling and downsampling operations, ensuring the maximal extraction of semantic information and the inheritance of the original features.

While the downsampling branch of the bidirectional pyramid incorporates the upsampled original data, the information passing through multiple convolution operations during the process can still lead to a partial loss of the original feature information in the backbone feature-extraction network. Training the model with downscaled original information, which leads to information loss, can result in biased learning of the data during the training process. As a consequence, the model may struggle to accurately capture the most representative defect features, thus affecting the precision of defect detection. The increased complexity of fusion strategies can lead to larger model sizes, which may impose higher demands on the running devices. Therefore, our main objective is to minimize the feature loss caused by convolution operations during the fusion process while ensuring computational efficiency within the given constraints of controlling computational costs. Through experiments and inference, we have observed that the BiFPN [29] can integrate more features without increasing computational costs. Its unique skip-connections structure also effectively addresses the issue of feature loss, aligning well with our desired improvement direction. Based on the aforementioned analysis, we introduce the BiFPN structure into the neck network, as shown in Figure 4. When the original input and output nodes of the neck network are in the same layer, we incorporate additional edges between them to minimize the loss of spatial information.

Considering the actual situation, replacing the original PANet with BiFPN is more effective in combining high-resolution and low-resolution image features for the dataset used in this study, where the images have relatively small dimensions. This is especially beneficial for identifying defects with fine details, such as scratches and rolled-in scales, which contain numerous small objects. The adoption of BiFPN leads to improved recognition performance for these types of defect images. To ensure data integrity and optimize computational cost, the original BiFPN assigns different weights to different input feature branches and performs element-wise multiplication. This process emphasizes the importance of each branch and allows for a reduction in the weight of less influential branches, thereby minimizing interference during the learning process. However, we observed that the introduction of applying weighted BiFPN to YOLOv7 did not yield the desired results and instead increased computational complexity. We believe that weighting different input feature layers and adding attention mechanisms are quite similar. The neck model already possesses strong learning capabilities, and introducing additional weight mechanisms may lead to an increase in trainable parameters, resulting in slower inference speed. Therefore, we removed the weight component of BiFPN and employed a de-weighted BiFPN instead. Through experiments, we observed that the mean average precision (mAP) increased from 81.4% to 83.4% compared to the original YOLOv7 neck network. Additionally, the parameter count experienced only a marginal increase, allowing us to maintain a fast inference speed.

In summary, this paper proposes an improved YOLOv7 neck network utilizing a de-weighted BiFPN for feature fusion. In the original neck network, we introduce an additional upsampling path that directly connects the backbone network with the downsampling branches. And we incorporate the original weights from the backbone network into the down-sampling branches to ensure proper convergence of the model during feature fusion. Additionally, this improvement carefully controls the increase in parameter count to maintain fast inference speed. Experimental results also validate the feasibility and effectiveness of our proposed approach.

## 4. Experiments

In the experiments, the hardware configuration consisted of an operating system Ubuntu 20.04 and a central processing unit (CPU) Intel Xeon(R) Gold 6330. GPU NVIDIA GeForce RTX 3090 was used in the setup. The software environment included Python 3.7.15, and the development framework used was Pytorch 1.13.0.

To ensure the comparability of the ablation experiments, it was essential to maintain consistent settings for each experiment in terms of epochs and hyperparameters. The batch size was set to 16, and the number of epochs was set to 300.

### 4.1. Datasets

In order to validate the effectiveness of the defect detection method proposed in this paper, we employed the NEU dataset to evaluate the algorithm’s accuracy, robustness, and generalization ability. The NEU-DET (Northeast Normal University) dataset [19] consists of images captured from real metal surfaces, covering various common metal defects. It is a commonly used computer vision dataset for material surface defect detection, created by researchers from Northeast Normal University in 2018. The dataset captures images of six distinct typical surface defects that arise during the manufacturing of steel plates due to various factors such as production, transportation, and storage. These defects include Rolled-in Scale (RS), Patches (Pa), Scratches (Cr), Pitted Surface (PS), Inclusions (In), and Scratches (Sc). Each type of defect contains 300 grayscale images with an original resolution of 200×200 pixels, resulting in a total of 1800 images. Subsequently, the collected images of defects undergo manual annotation, where the actual positions and sizes of the defects are labeled. This annotated information will provide significant assistance for both training and evaluating the algorithm.

The sample images in the dataset undergo data preprocessing and augmentation before being fed into the detection network. Specific operations include adjusting image dimensions, manipulating image colors, and modifying image shapes, among other dimensions. The adjustment of image dimensions employs methods such as mosaic and random resizing. These techniques are employed to enrich the diversity of sizes of training images. Color adjustments are applied using the Hue, Saturation, Value (HSV) color space for random modifications of saturation, brightness, and contrast. This simulates varying lighting and environmental conditions, thereby enhancing the model’s robustness and generalization ability. Additionally, images are subject to random flipping, random cropping, random translation, and random scaling. These augmentations simulate different sampling angles and aid the model in adapting to various object arrangements.

The anchor boxes of the YOLO-V7 algorithm are obtained through K-means clustering on the COCO (Microsoft Common Objects in Context) dataset. However, due to the significant difference in data distribution between the COCO dataset and the NEU dataset, we reclustered the anchor boxes using the K-means++ algorithm. The K-means++ algorithm is an improved version of the K-means algorithm. It enhances the stability and clustering performance of the K-means algorithm by improving the process of selecting initial cluster centers. K-means++ aims to avoid getting stuck in local optima and reduces dependency on the initial cluster center selection. In comparison to the random selection of initial cluster centers, the K-means++ algorithm typically achieves better clustering results. The anchor points obtained through K-means++ clustering with K=9 on the NEU-DET dataset are as follows: [62, 127], [77, 255], [161, 157], [512, 82], [205, 269], [104, 538], [438, 213], [227, 494], [536, 550].

### 4.2. Evaluations Metrics

In the field of defect detection, choosing appropriate evaluation metrics is crucial for assessing algorithm performance. The selected evaluation metrics should objectively measure the accuracy and robustness of the algorithm. In practical industrial production, both the accuracy and speed of defect detection are particularly important. If the recognition accuracy of defects is too low during the detection process, the machine is prone to making incorrect judgments and cannot accurately identify defective workpieces with quality issues. The steel-strip defect detection model is evaluated using Average Precision (AP), mean Average Precision (mAP), and Frames Per Second (FPS). AP represents the average detection precision for each defect category. mAP represents the average detection precision across all defect categories. It comprehensively considers both the accuracy and recall of the detection results. In the context of defect detection tasks, mAP provides insights into the algorithm’s precision and recall when detecting defects, i.e., the ratio between the detected defects and the actual number of defects present. FPS is a metric that measures the speed of algorithm processing and model inference time, representing the number of image frames processed per second. A higher FPS (shorter inference time) usually indicates lower computational complexity, implying that the algorithm can complete computations in a relatively shorter time, enabling real-time detection.
(8)$$Precision=\frac{TP}{TP+FP}$$
(9)$$Recall=\frac{TP}{TP+FN}$$
(10)$$mAP=\frac{\sum_{i=1}^{c}\int_{0}^{1}P(R)dR}{c}$$
(11)$$FPS=\frac{1}{AverageProcessingTime}$$

In the evaluation metrics, TP represents the number of correctly detected defect samples; FP represents the number of non-defect samples falsely detected as defects; and FN represents the number of defect samples that were not detected correctly. *P* and *R*, respectively, denote precision (the ratio of TP to the sum of TP and FP) and recall (the ratio of TP to the sum of TP and FN). The average processing time is a metric that measures the average time required for an algorithm to process a single frame of an image. It typically consists of the sum of the model inference time and the non-maximum suppression (NMS) time. Model inference time refers to the time it takes to input an image into the model, perform forward inference, and obtain the output.

### 4.3. Ablation Study

In this section, we conduct ablation studies to validate the effectiveness of our proposed improvements. Specifically, we investigate the impact of the modifications made to the main backbone network and the neck of the network, as well as the inclusion of the improved attention mechanism. The results of these experiments provide evidence for the effectiveness of each component in enhancing the overall performance of the model. The results of the ablation study are displayed in Table 1.

Backbone network: By incorporating the Fusion-Faster module to improve the YOLOv7 main feature extraction network, the model parameters decreased from 37.22 M to 31.89 M, and the corresponding Giga Floating Point Operations (GFLOPs) decreased from 105.2 to 39.1. At the same time, the mAP improved from 81.36 to 84.39, while the FPS remained relatively unchanged. This indicates that integrating the Fusion-Faster module into the backbone feature-extraction network can enhance the model’s detection accuracy and reduce the model’s parameter count without sacrificing speed.

Attention mechanism module: By inserting the proposed SCA attention module at the end of the feature-extraction network, the aim is to enhance the sensitivity of different-scale targets to position information. Comparing the data in rows 1 and 3 of the comparison table, it is evident that adding the SCA attention module before upsampling can emphasize the correlation of position information by introducing a small number of parameters, leading to improved model performance. The model parameters only increased from 37.22 M to 37.34 M. In contrast, the mAP improved from 81.36% to 83.94%, confirming the effectiveness of the SCA attention module. And the SCA module can be inserted at different locations within the network. We conducted experiments by placing the SCA module in deeper layers of the feature-extraction network. While it led to an improvement in model accuracy, the gains in accuracy were not as significant as when adding the attention mechanism to the end of the main network. Additionally, inserting the SCA module in shallow-layers resulted in increased computation due to a larger number of channels, leading to a noticeable decrease in model inference speed. After carefully considering the trade-off between speed and accuracy, we ultimately decided to place the SCA module at the end of the main network.

Neck network: A comparison between the first row and the fourth row in Table 1 clearly demonstrates the improvement in accuracy achieved by replacing the original neck network of YOLOv7 with the proposed modified version using the de-weighted BiFPN. Due to the well designed structure of BiFPN, it efficiently fuses multi-scale features before downsampling, resulting in improved detection accuracy. The mAP has increased from 81.36% to 83.39% without causing notable impacts on the model’s inference speed or size. With the incorporation of the improved backbone network, the model’s detection accuracy, as measured by mAP, has further reached 84.41%. Remarkably, the inference speed remains stable at around 70 FPS, demonstrating the successful synergy of the enhanced backbone and the efficient architecture, leading to an overall high performance defect detection system.

### 4.4. Comparison of Different Defect Detection Algorithms

In order to assess the effectiveness of our proposed model, we conducted comparative experiments with several mainstream methods, as shown in Table 2. The results of LFF-YOLO and Improved MSFT-YOLO are from the paper [38,45]. The results of YOLO-V3-based and RDD-YOLO are from the paper [5,36]. According to Table 2, our proposed model achieved the highest mAP compared to other methods, with an increment of 4.5% over YOLOv7. Furthermore, our model reduced 66 GFLOPs, resulting in a weight reduction of 62.7%. Despite these improvements, the model’s running speed remained at 70 FPS, demonstrating excellent real-time detection performance. EFC-YOLO achieved an mAP of 85.9%, surpassing YOLOv5m by 7.4% in mAP and outperforming YOLOv4m-mish by 5.6% in mAP. Compared to RDD-YOLO, EFC-YOLO achieved a 4.8% increase in mAP while maintaining a slightly faster speed. In comparison with MSFT-YOLO, we demonstrated significant advantages in both speed and accuracy. Although LFF-YOLO’s use of ShuffleNetv2 as the feature-extraction network demonstrated impressive GFLOPs reduction, with only 6.85 GFLOPs, our model maintained a superiority in mAP (85.9% compared to 79.2%). This highlights the effectiveness of our model in achieving a balance between computational efficiency and detection accuracy, making it a strong contender in the field of object detection. Moreover, when compared to the latest YOLOv8m, EFC-YOLO improved mAP from 80.1% to 85.9%. The use of a single-stage model in combination with the Fusion-Faster module effectively mitigated the computational overhead caused by attention mechanisms, ensuring the model’s efficient running speed. This advantage is also evident in Table 1. The experimental results and analysis presented above provide strong evidence to demonstrate the capabilities of our model in detecting surface defects on steel materials. With its superior performance compared to other state-of-the-art methods, the proposed model stands as a promising solution for accurate and efficient defect detection on steel surfaces, contributing to enhanced quality control and improved manufacturing processes.

Furthermore, we selected several state-of-the-art attention mechanisms and conducted comparative experiments on the NEU-DET dataset to assess the impact of different attention mechanisms on the performance of our proposed model. We compared the effects of different attention mechanisms on both the detection speed and performance of the model in Table 3.

In Figure 5, the performance of our model EFC-YOLO (on the right) on the NEU dataset is illustrated, showcasing the predicted bounding boxes, defect categories, and confidence scores. Compared to the baseline YOLOv7 (on the left), our model successfully identifies a greater number of subtle and minor defects, highlighting its superior capability in defect detection.

## 5. Discussion

However, there is still room for optimization in the proposed algorithm in this study. The CA attention mechanism captures position information more sensitively, enhances long-range dependencies in the information, and improves the accuracy of the model. Although attempting to incorporate the weight reduction improved the attention module at different positions in the network to explore the optimal location for module integration and network optimization, the results were not satisfactory. The usage of the attention module still introduced a computational burden to the lightweight adjusted network, thereby impacting the detection speed to a certain extent. Our future research direction will start with model-distillation methods to further compress the model. The goal is to enable better deployment of the model on mobile terminals and achieve industrial-scale usage while maintaining its performance and accuracy. On the other hand, the ability of neural networks to extract meaningful features from the input images can be hindered when the input images contain more noise or when the contrast of target defects is too low. Noise and low-contrast issues may cause the neural network to be more inclined to fit the noise rather than the actual features, which can negatively impact the model’s fitting accuracy and detection precision. To address the aforementioned issues, our future direction will focus on data augmentation and diverse data preprocessing techniques to generate a more varied set of samples, so that the model has higher robustness in the face of complex features.

## 6. Conclusions

This paper proposes a novel steel-surface-defect detector named EFC-YOLO, which is based on the design of YOLOv7. The detector has been improved in both the backbone and neck modules and incorporates attention mechanisms, resulting in excellent performance in terms of detection accuracy and speed. To enhance feature-extraction capabilities and reduce model size, we adopt the Fusion-Faster module as the primary component of the EFC-YOLO backbone. The utilization of PConv significantly reduces the model size while ensuring high detection accuracy. In the neck part of the network, the BiFPN aims to deepen the network and integrate features from different scales. By employing a more efficient information propagation mechanism, it retains a greater amount of original information. By incorporating the SCA attention module before the final upsampling operation in the backbone network, we have achieved significant improvements in handling long-range dependencies and better feature fusion. To validate the algorithm’s robustness and generalization, experiments were conducted on the NEU-DET dataset. Compared to state-of-the-art methods, EFC-YOLO achieved a mAP of 85.9% and a running speed of 70 FPS on NEU-DET. In comparison to the original YOLOv7 algorithm, our model’s GFLOP was reduced by 62.8% through lightweight improvements, resulting in a final GFLOP of 39.2. The experimental results demonstrate that EFC-YOLO can meet the requirements for real-time detection in terms of both accuracy and speed. Additionally, it has achieved a certain level of lightweight optimization.

## Figures and Tables

**Figure 1 sensors-23-07619-f001:**
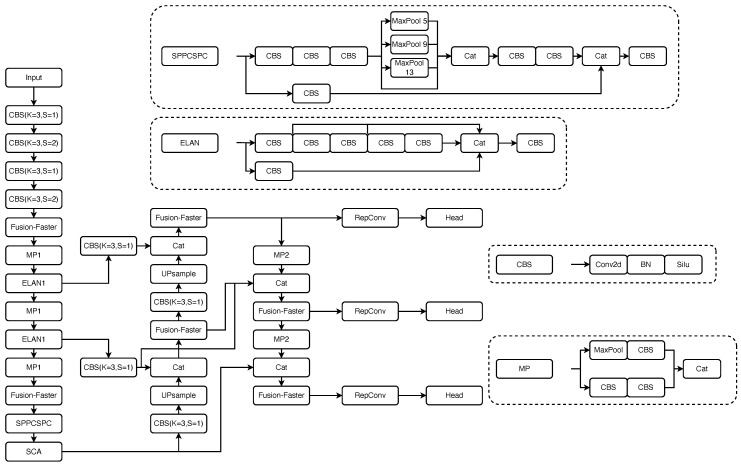
The architecture of EFC-YOLO.

**Figure 2 sensors-23-07619-f002:**
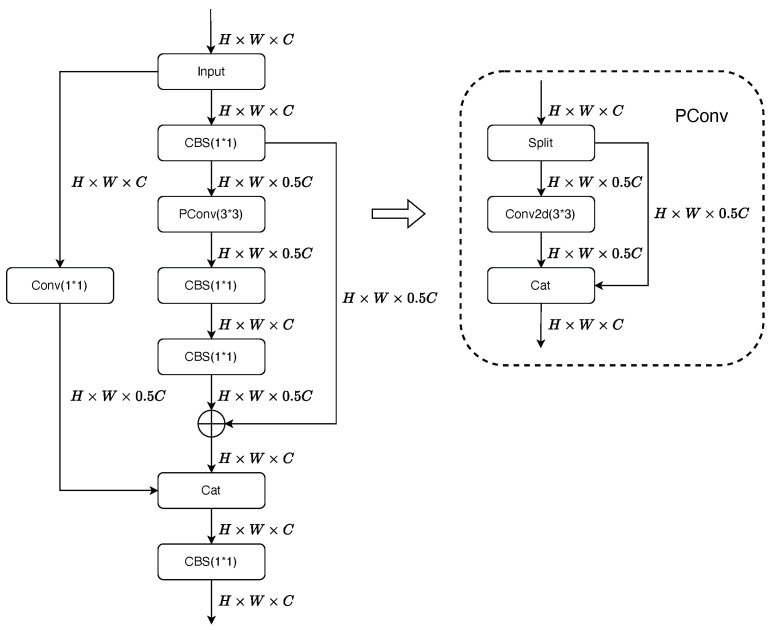
The architecture of Fusion-Faster module and PConv.

**Figure 3 sensors-23-07619-f003:**
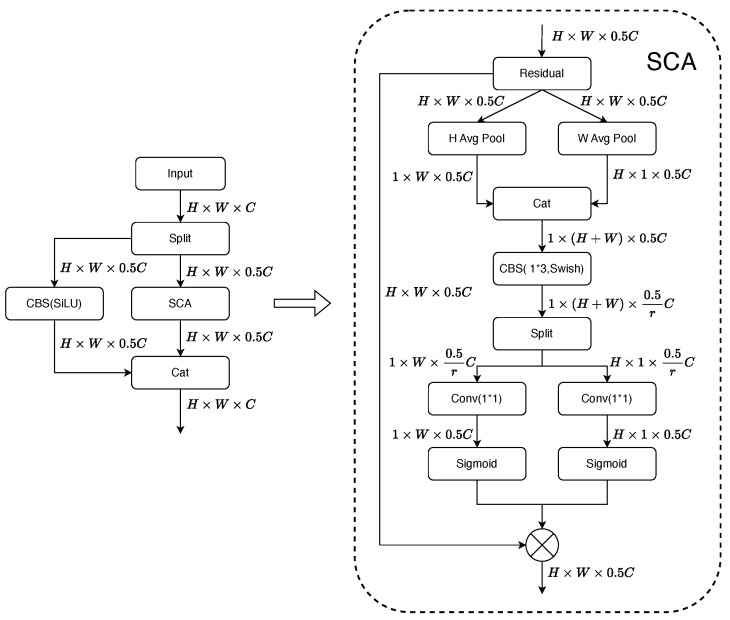
The architecture of Shortcut Coordinate Attention module.

**Figure 4 sensors-23-07619-f004:**
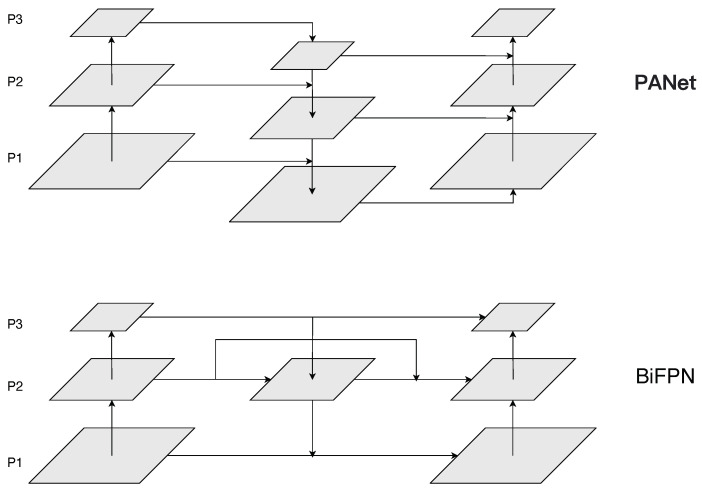
The architecture of BiFPN and PANet.

**Figure 5 sensors-23-07619-f005:**
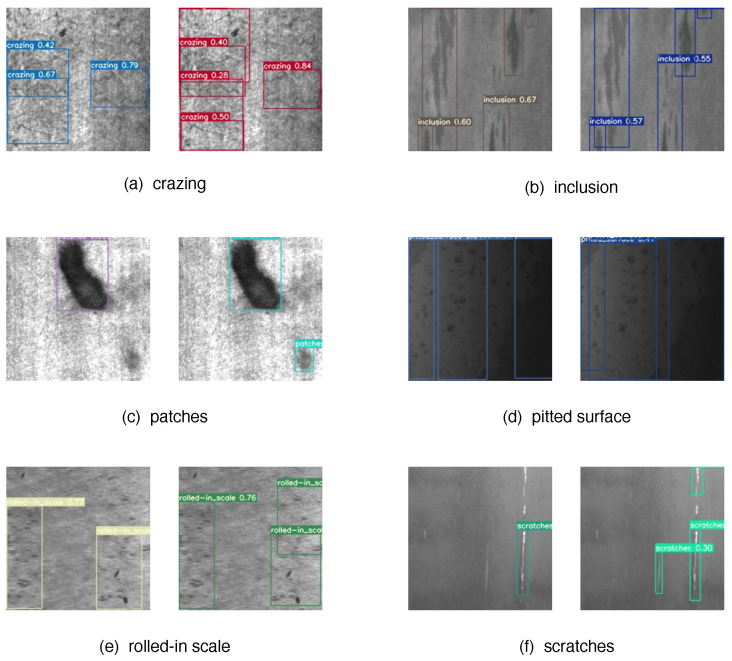
Results of YOLOv7 and EFC-YOLO on NEU-DET.

**Table 1 sensors-23-07619-t001:** Impact of changes in the YOLOv7 architecture evaluated on NEU-DET. (“V7” represents YOLOv7, “FF” represents Fusion-Faster module, “Bi” represents BiFPN, “SCA” represents the Shortcut Coordinate Attention module).

Change	map@0.5	GFLOPs	Params/M	FPS
V7	81.36%	105.2	37.22	76.3
V7 + FF	84.39%	39.1	31.74	87.3
V7 + SCA	83.94%	105.3	37.34	62.8
V7 + Bi	83.39%	105.6	37.35	83.8
V7 + FF + SCA	84.94%	39.1	31.82	68.3
V7 + FF + Bi	84.41%	39.1	31.78	90.5
V7 + SCA + Bi	83.13%	105.7	37.47	58.8
V7 + FF + SCA + Bi	85.86%	39.2	31.89	73.4

**Table 2 sensors-23-07619-t002:** Performance of the EFC-YOLO algorithm and other object detection algorithms.

Methods	map@0.5	GFLOPs	Params/M	FPS
YOLOv4m-mish	80.36%	53.4	24.4	55.2
YOLOv5m	78.44%	48.3	20.9	60.7
YOLOv7	81.36%	105.2	35.5	76.3
YOLOv8m	80.11%	78.7	28.5	82.8
LFF-YOLO [38]	79.23%	6.85	60.51	63.3
MSFT-YOLO [45]	75.20%	-	90.80	29.1
YOLO-V3-based model [5]	72.2%	-	-	64.5
RDD-YOLO [36]	81.1%	-	-	57.8
EFC-YOLO	85.86%	39.2	30.4	73.4

**Table 3 sensors-23-07619-t003:** Performance of varying attention-mechanism modules.

Change	map@0.5	GFLOPs	Params/M	FPS
CA	84.83%	39.2	30.3	62.4
ECA	84.27%	39.1	30.2	79.5
SimAM	84.66 %	39.1	30.1	65.8
CBAM	84.73%	39.2	30.3	63.8
GAM	84.02%	39.5	31.9	66.3
SCA	85.86%	39.2	39.1	73.4

## Data Availability

NEU-DET (Northeast Normal University) dataset: http://faculty.neu.edu.cn/songkechen/zh_CN/zdylm/263270/list/index.htm (accessed on 22 March 2023).

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
