# Peer review of "EFC-YOLO: An Efficient Surface-Defect-Detection Algorithm for Steel Strips"

_sensors, 2023, doi:10.3390/s23177619_

Round 1

Reviewer 1 Report

Overall a good piece of research, focused on automated steel surface defect detection. 

For improvement;

There are many YOLO variants, why YOLOv7 variant was selected. To answer this you could provide a comparison of works using YOLOv7 for similar works such as pallet racking inspection.

Also try to mention some reviews on YOLO variants i.e., 'YOLOv1-YOLOv8 industrial  review', this will direct readers towards complementary review papers to better understand the inner workings of different variants before making a selection.

Overall Good Work.

Good.

Reviewer 2 Report

I am honored to review this paper as an expert. This paper proposes a new method , EFC-YOLO , for efficient surface defect detection. From the results, it had a very high [email protected]  value, which was higher than other algorithms. Although the source of the dataset was mentioned in the article, I can not see the specific data and the source code of the new method. If the source code can be placed on Github or other places to facilitate open source testing, I can give the advice of accept.

Please find a professional polishing company to optimize the language of the paper.

Reviewer 3 Report

1.     The different abbreviations used by the authors in the manuscript should be expanded when they first appear in the manuscript.

2.     How authors have dealt with overfitting and underfitting?

3.     The robust comparison with other classifiers is also be incorporated in the manuscript.

4.     How data has been acquired? The complete procedure should also be incorporated in the manuscript.

5.     The computational complexity should also be incorporated in the manuscript.

NA

Round 2

Reviewer 3 Report

1.      The computational complexity of the proposed work is still incomplete within the manuscript. Also, it can be shown by the time taken by the proposed algorithm. 
